# Learning a Concept Hierarchy from Multi-labeled Documents

**Viet-An Nguyen**[1,*]**Jordan Boyd-Graber**[2], **Philip Resnik**[1,3,4], **Jonathan Chang**[5]
[1]Computer Science, [3]Linguistics, [4]UMIACS     [2]Computer Science     [5]Facebook
Univ. of Maryland, College Park, MD    Univ. of Colorado, Boulder, CO    Menlo Park, CA
vietan@cs.umd.edu        Jordan.Boyd.Graber    jonchang@fb.com
resnik@umd.edu           @colorado.edu

## Abstract

While topic models can discover patterns of word usage in large corpora, it is difficult to meld this unsupervised structure with noisy, human-provided labels, especially when the label space is large. In this paper, we present a model—Label to Hierarchy (L2H)—that can induce a hierarchy of user-generated labels and the topics associated with those labels from a set of multi-labeled documents. The model is robust enough to account for missing labels from untrained, disparate annotators and provide an interpretable summary of an otherwise unwieldy label set. We show empirically the effectiveness of L2H in predicting held-out words and labels for unseen documents.

## 1   Understanding Large Text Corpora through Label Annotations

Probabilistic topic models [4] discover the thematic structure of documents from news, blogs, and web pages. Typical unsupervised topic models such as latent Dirichlet allocation [7, LDA] uncover topics from unannotated documents. In many settings, however, documents are also associated with additional data, which provide a foundation for joint models of text with continuous response variables [6, 48, 27], categorical labels [37, 18, 46, 26] or link structure [9].

This paper focuses on additional information in the form of *multi-labeled data*, where each document is tagged with a set of labels. These data are ubiquitous. Web pages are tagged with multiple directories,[1] books are labeled with different categories or political speeches are annotated with multiple issues.[2] Previous topic models on multi-labeled data focus on a small set of relatively independent labels [25, 36, 46]. Unfortunately, in many real-world examples, the number of labels— from hundreds to thousands—is incompatible with the independence assumptions of these models.

In this paper, we capture the dependence among the labels using a learned tree-structured hierarchy. Our proposed model, L2H—*Label to Hierarchy*—learns from label co-occurrence and word usage to discover a hierarchy of topics associated with user-generated labels. We show empirically that L2H can improve over relevant baselines in predicting words or missing labels in two prediction tasks. L2H is designed to explicitly capture the *relationships* among labels to discover a highly interpretable hierarchy from multi-labeled data. This interpretable hierarchy helps improve prediction performance and also provides an effective way to search, browse and understand multi-labeled data [17, 10, 8, 12].

---

[*]Part of this work was done while the first author interned at Facebook.
[1]Open Directory Project (http://www.dmoz.org/)
[2]Policy Agenda Codebook (http://policyagendas.org/)

# 2 L2H: Capturing Label Dependencies using a Tree-structured Hierarchy

Discovering a topical hierarchy from text has been the focus of much topic modeling research. One popular approach is to learn an *unsupervised* hierarchy of topics. For example, hLDA [5] learns an unbounded tree-structured hierarchy of topics from unannotated documents. One drawback of hLDA is that documents only are associated with a single path in the topic tree. Recent work relaxing this restriction include TSSB [1], nHDP [30], nCRF [2] and SHLDA [27]. Going beyond tree structure, PAM [20] captures the topic hierarchy using a pre-defined DAG, inspiring more flexible extensions [19, 24]. However, since only unannotated text is used to infer the hierarchical topics, it usually requires an additional step of topic labeling to make them interpretable. This difficulty motivates work leveraging existing taxonomies such as HSLDA [31] and hLLDA [32].

A second active area of research is constructing a taxonomy from multi-labeled data. For example, Heymann and Garcia-Molina [17] extract a tag hierarchy using the tag network centrality; similar work has been applied to protein hierarchies [42]. Hierarchies of concepts have come from seeded ontologies [39], crowdsourcing [29], and user-specified relations [33]. More sophisticated approaches build domain-specific keyword taxonomies with adapting Bayesian Rose Trees [21]. These approaches, however, concentrate on the tags, ignoring the *content* the tags describe.

In this paper, we combine ideas from these two lines of research and introduce L2H, a hierarchical topic model that discovers a tree-structured hierarchy of concepts from a collection of multi-labeled documents. L2H takes as input a set of $D$ documents $\{\boldsymbol{w}_d\}$, each tagged with a set of labels $\boldsymbol{l}_d$. The label set $\mathcal{L}$ contains $K$ unique, unstructured labels and the word vocabulary size is $V$. To learn an interpretable taxonomy, L2H associates each *label*—a user-generated word/phrase—with a *topic*—a multinomial distribution over the vocabulary—to form a *concept*, and infers a tree-structured hierarchy to capture the relationships among concepts. Figure 1 shows the plate diagram for L2H, together with its generative process.

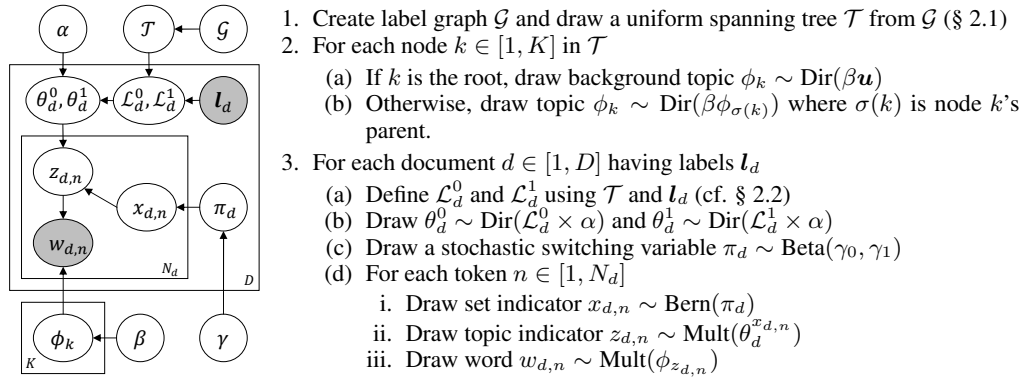

1. Create label graph $\mathcal{G}$ and draw a uniform spanning tree $\mathcal{T}$ from $\mathcal{G}$ (§ 2.1)
2. For each node $k \in [1, K]$ in $\mathcal{T}$
   (a) If $k$ is the root, draw background topic $\phi_k \sim \text{Dir}(\beta \boldsymbol{u})$
   (b) Otherwise, draw topic $\phi_k \sim \text{Dir}(\beta \phi_{\sigma(k)})$ where $\sigma(k)$ is node $k$'s parent.
3. For each document $d \in [1, D]$ having labels $\boldsymbol{l}_d$
   (a) Define $\mathcal{L}_d^0$ and $\mathcal{L}_d^1$ using $\mathcal{T}$ and $\boldsymbol{l}_d$ (cf. § 2.2)
   (b) Draw $\theta_d^0 \sim \text{Dir}(\mathcal{L}_d^0 \times \alpha)$ and $\theta_d^1 \sim \text{Dir}(\mathcal{L}_d^1 \times \alpha)$
   (c) Draw a stochastic switching variable $\pi_d \sim \text{Beta}(\gamma_0, \gamma_1)$
   (d) For each token $n \in [1, N_d]$
      i. Draw set indicator $x_{d,n} \sim \text{Bern}(\pi_d)$
      ii. Draw topic indicator $z_{d,n} \sim \text{Mult}(\theta_d^{x_{d,n}})$
      iii. Draw word $w_{d,n} \sim \text{Mult}(\phi_{z_{d,n}})$

Figure 1: Generative process and the plate diagram notation of L2H.

## 2.1 Generating a labeled topic hierarchy

We assume an underlying directed graph $\mathcal{G} = (\mathcal{E}, \mathcal{V})$ in which each node is a concept consisting of (1) a label—observable user-generated input, and (2) a topic—latent multinomial distribution over words.[3] The prior weight of a directed edge from node $i$ to node $k$ is the fraction of documents tagged with label $k$ which are also tagged with label $i$: $t_{i,k} = D_{i,k}/D_j$. We also assume an additional Background node. Edges to the Background node have prior zero weight and edges from the Background node to node $i$ have prior weight $t_{\text{root},i} = D_i/\max_k D_k$. Here, $D_i$ is the number of documents tagged with label $i$, and $D_{i,k}$ is the number of documents tagged with both labels $i$ and $k$.

The tree $\mathcal{T}$ is a spanning tree generated from $\mathcal{G}$. The probability of a tree given the graph $\mathcal{G}$ is thus the product of all its edge prior weights $p(\mathcal{T} \mid \mathcal{G}) = \prod_{e \in \mathcal{E}} t_e$. To capture the intuition that child nodes in the hierarchy specialize the concepts of their parents, we model the topic $\phi_k$ at each node

$k$ using a Dirichlet distribution whose mean is centered at the topic of node $k$'s parent $\sigma(k)$, i.e., $\phi_k \sim \text{Dir}(\beta\phi_{\sigma(k)})$. The topic at the root node is drawn from a symmetric Dirichlet $\phi_{\text{root}} \sim \text{Dir}(\beta\boldsymbol{u})$, where $\boldsymbol{u}$ is a uniform distribution over the vocabulary [1, 2]. This is similar to the idea of "backoff" in language models where more specific contexts inherit the ideas expressed in more general contexts; i.e., if we talk about "pedagogy" in education, there's a high likelihood we'll also talk about it in university education [22, 41].

## 2.2 Generating documents

As in LDA, each word in a document is generated by one of the latent topics. L2H, however, also uses the labels and topic hierarchy to restrict the topics a document uses. The document's label set $\boldsymbol{l}_d$ identifies which nodes are more likely to be used. *Restricting* tokens of a document in this way—to be generated only from a subset of the topics depending the document's labels—creates specific, focused, labeled topics [36, Labeled LDA].

Unfortunately, $\boldsymbol{l}_d$ is unlikely to be an exhaustive enumeration: particularly when the label set is large, users often forget or overlook relevant labels. We therefore depend on the learned topology of the hierarchy to *fill in* the gaps of what users forget by expanding $\boldsymbol{l}_d$ into a broader set, $\mathcal{L}_d^1$, which is the union of nodes on the paths from the root node to any of the document's label nodes. We call this the document's *candidate set*. The candidate set also induces a *complementary set* $\mathcal{L}_d^0 \equiv \mathcal{L} \setminus \mathcal{L}_d^1$ (illustrated in Figure 2). Previous approaches such as LPAM [3] and Tree labeled LDA [40] also leverage the label hierarchy to expand the original label set. However, these previous models require that the label hierarchy is given rather than inferred as in our L2H.

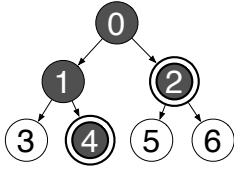

Figure 2: Illustration of the candidate label set: given a document $d$ having labels $\boldsymbol{l}_d = \{2, 4\}$ (double-circled nodes), the candidate label set of $d$ consists of nodes on all the paths from the root node to node 2 and node 4. $\mathcal{L}_d^1 = \{0, 1, 2, 4\}$ and $\mathcal{L}_d^0 = \{3, 5, 6\}$. This allows an imperfect label set to induce topics that the document *should* be associated with even if they weren't explicitly enumerated.

L2H replaces Labeled LDA's absolute *restriction* to specific topics to a soft *preference*. To achieve this, each document $d$ has a *switching variable* $\pi_d$ drawn from $\text{Beta}(\gamma_0, \gamma_1)$, which effectively decides how likely tokens in $d$ are to be generated from $\mathcal{L}_d^1$ versus $\mathcal{L}_d^0$. Token $n$ in document $d$ is generated by first flipping the biased coin $\pi_d$ to choose the set indicator $x_{d,n}$. Given $x_{d,n}$, the label $z_{d,n}$ is drawn from the corresponding label distribution $\theta_d^{x_{d,n}}$ and the token is generated from the corresponding topic $w_{d,n} \sim \text{Mult}(\phi_{z_{d,n}})$.

# 3 Posterior Inference

Given a set of documents with observed words $\{\boldsymbol{w}_d\}$ and labels $\{\boldsymbol{l}_d\}$, inference finds the posterior distribution over the latent variables. We use a Markov chain Monte Carlo (MCMC) algorithm to perform posterior inference, in which each iteration after initialization consists of the following steps: (1) sample a set indicator $x_{d,n}$ and topic assignment $z_{d,n}$ for each token, (2) sample a word distribution $\phi_k$ for each node $k$ in the tree, and (3) update the structure of the label tree.

**Initialization:** With the large number of labels, the space of hierarchical structures that MCMC needs to explore is huge. Initializing the tree-structure hierarchy is crucial to help the sampler focus on more important regions of the search space and help the sampler converge. We initialize the hierarchy with the maximum *a priori* probability tree by running Chu-Liu/Edmonds' algorithm to find the maximum spanning tree on the graph $\mathcal{G}$ starting at the background node.

**Sampling assignments $x_{d,n}$ and $z_{d,n}$:** For each token, we need to sample whether it was generated from the label set or not, $x_{d,n}$. We choose label set $i$ with probability $\frac{C_{d,i}^{-d,n}+\gamma_i}{C_{d,\cdot}^{-d,n}+\gamma_0+\gamma_1}$ and we sample a node in the chosen set $i$ with probability $\frac{N_{d,k}^{-d,n}+\alpha}{C_{d,i}^{-d,n}+\alpha|\mathcal{L}_d^i|} \cdot \phi_{k,w_{d,n}}$. Here, $C_{d,i}$ is the number of times tokens in document $d$ are assigned to label set $i$; $N_{d,k}$ is the number of times tokens in document $d$

are assigned to node $k$. Marginal counts are denoted by $\cdot$, and $^{-d,n}$ denotes the counts excluding the assignment of token $w_{d,n}$.

After we have the label set, we can sample the topic assignment. This is more efficient than sampling jointly, as most tokens are in the label set, and there are a limited number of topics in the label set. The probability of assigning node $k$ to $z_{d,n}$ is

$$p(x_{d,n} = i, z_{d,n} = k \mid \boldsymbol{x}^{-d,n}, \boldsymbol{z}^{-d,n}, \phi, \mathcal{L}_d^i) \propto \frac{C_{d,i}^{-d,n} + \gamma_i}{C_{d,\cdot}^{-d,n} + \gamma_0 + \gamma_1} \cdot \frac{N_{d,k}^{-d,n} + \alpha}{C_{d,i}^{-d,n} + \alpha|\mathcal{L}_d^i|} \cdot \phi_{k,w_{d,n}} \quad (1)$$

**Sampling topics $\phi$:**  As discussed in Section 2.1, topics on each path in the hierarchy form a cascaded Dirichlet-multinomial chain where the multinomial $\phi_k$ at node $k$ is drawn from a Dirichlet distribution with the mean vector being the topic $\phi_{\sigma(k)}$ at the parent node $\sigma(k)$. Given assignments of tokens to nodes, we need to determine the conditional probability of a word given the token. This can be done efficiently in two steps: bottom-up smoothing and top-down sampling [2].

- Bottom-up smoothing: This step estimates the counts $\tilde{M}_{k,v}$ of node $k$ propagated from its children. This can be approximated efficiently by using the minimal/maximal path assumption [11, 44]. For the minimal path assumption, each child node $k'$ of $k$ propagates a value of 1 to $\tilde{M}_{k,v}$ if $M_{k',v} > 0$. For the maximal path assumption, each child node $k'$ of $k$ propagates the full count $M_{k',v}$ to $\tilde{M}_{k,v}$.

- Top-down sampling: After estimating $\tilde{M}_{k,v}$ for each node from leaf to root, we sample the word distributions top-down using its actual counts $\boldsymbol{m}_k$, its children's propagated counts $\tilde{\boldsymbol{m}}_k$ and its parent's word distribution $\phi_{\sigma(k)}$ as $\phi_k \sim \text{Dir}(\boldsymbol{m}_k + \tilde{\boldsymbol{m}}_k + \beta\phi_{\sigma(k)})$.

**Updating tree structure $\mathcal{T}$:**  We update the tree structure by looping through each non-root node, proposing a new parent node and either accepting or rejecting the proposed parent using the Metropolis-Hastings algorithm. More specifically, given a non-root node $k$ with current parent $i$, we propose a new parent node $j$ by sampling from all incoming nodes of $k$ in graph $\mathcal{G}$, with probability proportional to the corresponding edge weights. If the proposed parent node $j$ is a descendant of $k$, we reject the proposal to avoid a cycle. If it is not a descendant, we accept the proposed move with probability $\min\left(1, \frac{Q(i \prec k)}{Q(j \prec k)} \frac{P(j \prec k)}{P(i \prec k)}\right)$, where $Q$ and $P$ denote the proposal distribution and the model's joint distribution respectively, and $i \prec k$ denotes the case where $i$ is the parent of $k$.

Since we sample the proposed parent using the edge weights, the proposal probability ratio is

$$\frac{Q(i \prec k)}{Q(j \prec k)} = \frac{t_{i,k}}{t_{j,k}} \quad (2)$$

The joint probability of L2H's observed and latent variables is:

$$P = \prod_{e \in \mathcal{E}} p(e \mid \mathcal{G}) \prod_{d=1}^{D} p(\boldsymbol{x}_d \mid \gamma) p(\boldsymbol{z}_d \mid \boldsymbol{x}_d, \boldsymbol{l}_d, \alpha) p(\boldsymbol{w}_d \mid \boldsymbol{z}_d, \phi) \prod_{l=1}^{K} p(\phi_l \mid \phi_{\sigma(l)}, \beta) p(\phi_{\text{root}} \mid \beta) \quad (3)$$

When node $k$ changes its parent from node $i$ to $j$, the candidate set $\mathcal{L}_d^1$ changes for any document $d$ that is tagged with any label in the subtree rooted at $k$. Let $\triangle_k$ denote the subtree rooted at $k$ and $\mathcal{D}_{\triangle_k} = \{d \mid \exists l \in \triangle_k \wedge l \in \boldsymbol{l}_d\}$ the set of documents whose candidate set might change when $k$'s parent changes. Canceling unchanged quantities, the ratio of the joint probabilities is:

$$\frac{P(j \prec k)}{P(i \prec k)} = \frac{t_{j,k}}{t_{i,k}} \prod_{d \in \mathcal{D}_{\triangle_k}} \frac{p(\boldsymbol{z}_d \mid j \prec k)}{p(\boldsymbol{z}_d \mid i \prec k)} \frac{p(\boldsymbol{x}_d \mid j \prec k)}{p(\boldsymbol{x}_d \mid i \prec k)} \frac{p(\boldsymbol{w}_d \mid j \prec k)}{p(\boldsymbol{w}_d \mid i \prec k)} \prod_{l=1}^{K} \frac{p(\phi_l \mid j \prec k)}{p(\phi_l \mid i \prec k)} \quad (4)$$

We now expand each factor in Equation 4. The probability of node assignments $\boldsymbol{z}_d$ for document $d$ is computed by integrating out the document-topic multinomials $\theta_d^0$ and $\theta_d^1$ (for the candidate set and its inverse):

$$p(\boldsymbol{z}_d \mid \boldsymbol{x}_d, \mathcal{L}_d^0, \mathcal{L}_d^1; \alpha) = \prod_{x \in \{0,1\}} \frac{\Gamma(\alpha|\mathcal{L}_d^x|)}{\Gamma(C_{d,x} + \alpha|\mathcal{L}_d^x|)} \prod_{l \in \mathcal{L}_d^x} \frac{\Gamma(N_{d,l} + \alpha)}{\Gamma(\alpha)} \quad (5)$$

Similarly, we compute the probability of $\boldsymbol{x}_d$ for each document $d$, integrating out $\pi_d$,

$$p(\boldsymbol{x}_d \,|\, \gamma) = \frac{\Gamma(\gamma_0 + \gamma_1)}{\Gamma(C_{d,\cdot} + \gamma_0 + \gamma_1)} \prod_{x \in \{0,1\}} \frac{\Gamma(C_{d,x} + \gamma_i)}{\Gamma(\gamma_x)} \qquad (6)$$

Since we explicitly sample the topic $\phi_l$ at each node $l$, we need to re-sample all topics for the case that $j$ is the parent of $i$ to compute the ratio $\prod_{l=1}^{K} \frac{p(\phi_l \,|\, j \prec k)}{p(\phi_l \,|\, i \prec k)}$. Given the sampled $\phi$, the word likelihood is $p(\boldsymbol{w}_d \,|\, \boldsymbol{z}_d, \phi) = \prod_{n=1}^{N_d} \phi_{z_{d,n}, w_{d,n}}$.

However, re-sampling the topics for the whole hierarchy for every node proposal is inefficient. To avoid that, we keep all $\phi$'s fixed and approximate the ratio as:

$$\prod_{d \in \mathcal{D}_{\triangle_k}} \frac{p(\boldsymbol{w}_d \,|\, j \prec k)}{p(\boldsymbol{w}_d \,|\, i \prec k)} \prod_{l=1}^{K} \frac{p(\phi_l \,|\, j \prec k)}{p(\phi_l \,|\, i \prec k)} \approx \frac{\int_{\phi_k} p(\boldsymbol{m}_k + \tilde{\boldsymbol{m}}_k \,|\, \phi_k) \, p(\phi_k \,|\, \phi_j) \, d\phi_k}{\int_{\phi_k} p(\boldsymbol{m}_k + \tilde{\boldsymbol{m}}_k \,|\, \phi_k) \, p(\phi_k \,|\, \phi_i) \, d\phi_k} \qquad (7)$$

where $\boldsymbol{m}_k$ is the word counts at node $k$ and $\tilde{\boldsymbol{m}}_k$ is the word counts propagated from children of $k$. Since $\phi$ is fixed and the node assignments $\boldsymbol{z}$ are unchanged, the word likelihoods cancel out except for tokens assigned at $k$ or any of its children. The integration in Equation 7 is

$$\int_{\phi_k} p(\boldsymbol{m}_k + \tilde{\boldsymbol{m}}_k \,|\, \phi_k) \, p(\phi_k \,|\, \phi_j) \, d\phi_k = \frac{\Gamma(\beta)}{\Gamma(M_{k,\cdot} + \tilde{M}_{k,\cdot} + \beta)} \prod_{v=1}^{V} \frac{\Gamma(M_{k,v} + \tilde{M}_{k,v} + \beta \phi_{i,v})}{\Gamma(\beta \phi_{i,v})} \quad (8)$$

Using Equations 2 and 4, we can compute the Metropolis-Hastings acceptance probability.

# 4 Experiments: Analyzing Political Agendas in U.S. Congresses

In our experiments, we focus on studying political attention in the legislative process, of interest to both computer scientists [13, 14] and political scientists [15, 34]. GovTrack provides bill text from the US Congress, each of which is assigned multiple political issues by the Congressional Research Service. Examples of Congressional issues include Education, Higher Education, Health, Medicare, etc. To evaluate the effectiveness of L2H, we evaluate on two computational tasks: document modeling—measuring perplexity on a held-out set of documents—and multi-label classification. We also discuss qualitative results based on the label hierarchy learned by our model.

**Data:** We use the text and labels from GovTrack for the 109[th] through 112[th] Congresses (2005–2012). For both quantitative tasks, we perform 5-fold cross-validation. For each fold, we perform standard pre-processing steps on the training set including tokenization, removing stopwords, stemming, adding bigrams, and filtering using TF-IDF to obtain a vocabulary of 10,000 words (final statistics in Figure 3).[4] After building the vocabulary from training documents, we discard all out-of-vocabulary words in the test documents. We ignore labels associated with fewer than 100 bills.

## 4.1 Document modeling

In the first quantitative experiment, we focus on the task of predicting the words in held-out test documents, given their labels. This is measured by *perplexity*, a widely-used evaluation metric [7, 45]. To compute perplexity, we follow the "*estimating $\theta$*" method described in Wallach et al. [45, Sec. 5.1] and split each test document $d$ into $\boldsymbol{w}_d^{\text{TE}_1}$ and $\boldsymbol{w}_d^{\text{TE}_2}$. During training, we estimate all topics' distributions over the vocabulary $\hat{\phi}$. During test, first we run Gibbs sampling using the learned topics on $\boldsymbol{w}_d^{\text{TE}_1}$ to estimate the topic proportions $\hat{\theta}_d^{\text{TE}}$ for each test document $d$. Then, we compute the perplexity on the held-out words $\boldsymbol{w}_d^{\text{TE}_2}$ as $\exp \left\{ -\frac{\sum_d \log\left(p(\boldsymbol{w}_d^{\text{TE}_2} \,|\, \boldsymbol{l}_d, \hat{\theta}_d^{\text{TE}}, \hat{\phi})\right)}{N^{\text{TE}_2}} \right\}$ where $N^{\text{TE}_2}$ is the total number of tokens in $\boldsymbol{w}_d^{\text{TE}_2}$.

**Setup** We compare our proposed model L2H with the following methods:

- LDA [7]: unsupervised topic model with a flat topic structure. In our experiments, we set the number of topics of LDA equal to the number of labels in each dataset.
- L-LDA [36]: associates each topic with a label, and a document is generated using the topics associated with the document's labels only.
- L2F (Label to Flat structure): a simplified version of L2H with a fixed, flat topic structure. The major difference between L2F and L-LDA is that L2F allows tokens to be drawn from topics that are not in the document's label set via the use of the switching variable (Section 2.2). Improvements of L2H over L2F show the importance of the *hierarchical* structure.

For all models, the number of topics is the number of labels in the dataset. We run for 1,000 iterations on the training data with a burn-in period of 500 iterations. After the burn-in period, we store ten sets of estimated parameters, one after every fifty iterations. During test time, we run ten chains using these ten learned models on the test data and compute the perplexity after 100 iterations. The perplexity of each fold is the average value over the ten chains [28].

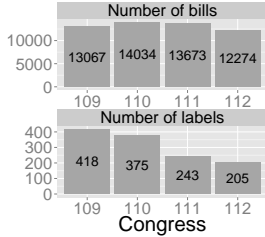 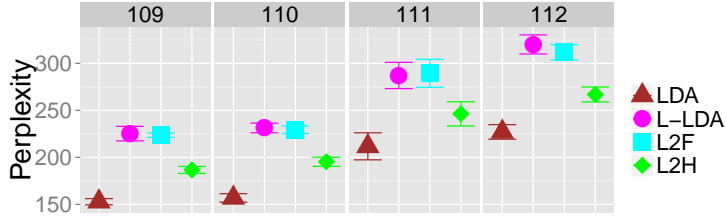

Figure 3: Dataset statistics    Figure 4: Perplexity on held-out documents, averaged over 5 folds

**Results:** Figure 4 shows the perplexity of the four models averaged over five folds on the four datasets. LDA outperforms the other models with labels since it can freely optimize the likelihood without additional constraints. L-LDA and L2F are comparable. However, L2H significantly outperforms both L-LDA and L2F. Thus, if incorporating labels into a model, learning an additional topic hierarchy improves predictive power and generalizability of L-LDA.

### 4.2 Multi-label Classification

Multi-label classification is predicting a set of labels for a test document given its text [43, 23, 47]. The prediction is from a set of pre-defined $K$ labels and each document can be tagged with any of the $2^K$ possible subsets. In this experiment, we use M3L—an efficient max-margin multi-label classifier [16]—to study how features extracted from our L2H improve classification.

We use $F_1$ as the evaluation metric. The $F_1$ score is first computed for each document $d$ as $F_1(d) = \frac{2 \cdot P(d) \cdot R(d)}{P(d) + R(d)}$, where $P(d)$ and $R(d)$ are the precision and recall for document $d$. After $F_1(d)$ is computed for all documents, the overall performance can be summarized by micro-averaging and macro-averaging to obtain Micro-$F_1$ and Macro-$F_1$ respectively. In macro-averaging, $F_1$ is first computed for each document using its own confusion matrix and then averaged. In micro-averaging, on the other hand, only a single confusion matrix is computed for all documents, and the $F_1$ score is computed based on this single confusion matrix [38].

**Setup** We use the following sets of features:

- TF: Each document is represented by a vector of term frequency of all word types in the vocabulary.
- TF-IDF: Each document is represented by a vector $\psi_d^{\text{TFIDF}}$ of TF-IDF of all word types.
- L-LDA&TF-IDF: Ramage et al. [35] combine L-LDA features and TF-IDF features to improve the performance on recommendation tasks. Likewise, we extract a $K$-dimensional vector $\hat{\theta}_d^{\text{L-LDA}}$ and combine with TF-IDF vector $\psi_d^{\text{TFIDF}}$ to form the feature vector of L-LDA&TF-IDF.[5]

- L2H&TF-IDF: Similarly, we combine TF-IDF with the features $\hat{\theta}_d^{\text{L2H}} = \{\hat{\theta}_d^0, \hat{\theta}_d^1\}$ extracted using L2H (same MCMC setup as L-LDA).

One complication for L2H is the candidate label set $\mathcal{L}_d^1$, which is not observed during test time. Thus, during test time, we estimate $\mathcal{L}_d^1$ using TF-IDF. Let $\mathcal{D}_l$ be the set of documents tagged with label $l$. For each $l$, we compute a TF-IDF vector $\phi_l^{\text{TFIDF}} = \text{avg}_{d \in \mathcal{D}_l} \psi_d^{\text{TFIDF}}$. Then for each document $d$, we generate the $k$ nearest labels using cosine similarity, and add them to the candidate label set $\mathcal{L}_d^1$ of $d$. Finally, we expand this initial set by adding all labels on the paths from the root of the learned hierarchy to any of the $k$ nearest labels (Figure 2). We explored different values of $k \in \{3, 5, 7, 9\}$, with similar results; the results in this section are reported with $k = 5$.

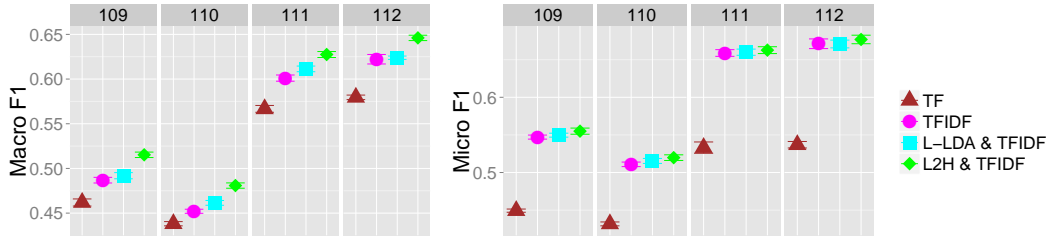

Figure 5: Multi-label classification results. The results are averaged over 5 folds.

**Results**  Figure 5 shows classification results. For both Macro-$F1$ and Micro-$F1$, TF-IDF, L-LDA&TF-IDF and L2H&TF-IDF significantly outperform TF. Also, L-LDA&TF-IDF performs better than TF-IDF, which is consistent with Ramage et al. (2010) [35].

L2H&TF-IDF performs better than L-LDA&TF-IDF, which in turn performs better than TF-IDF. This shows that features extracted from L2H are more predictive than those extracted from L-LDA, and both improve classification. The improvements of L2H&TF-IDF and L-LDA&TF-IDF over TF-IDF are clearer for Macro-$F1$ compared with Micro-$F1$. Thus, features from both topic models help improve prediction, regardless of the frequencies of their tagged labels.

### 4.3 Learned label hierarchy: A taxonomy of Congressional issues

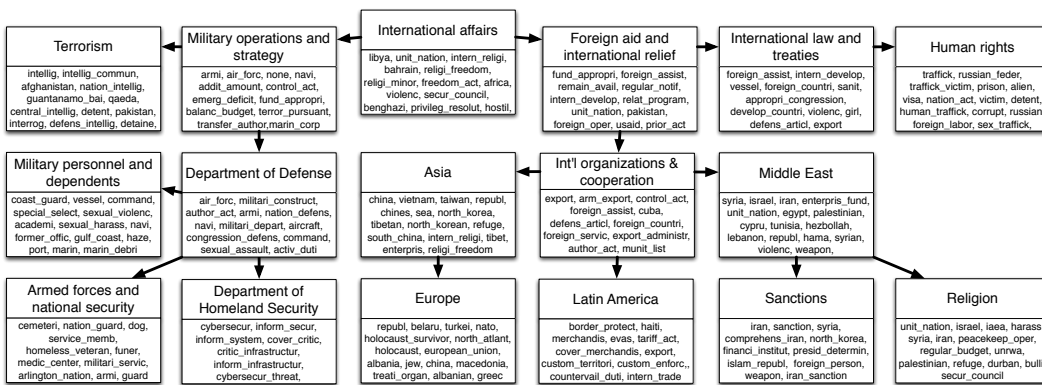

Figure 6: A subtree in the hierarchy learned by L2H. The subtree root International Affairs is a child node of the Background root node.

To qualitatively analyze the hierarchy learned by our model, Figure 6 shows a subtree whose root is about International Affairs, obtained by running L2H on bills in the $112^{th}$ U.S. Congress. The learned topic at International Affairs shows the focus of $112^{th}$ Congress on the Arab Spring—a revolutionary wave of demonstrations and protests in Arab countries like Libya, Bahrain, etc. The concept is then split into two distinctive aspects of international affairs: Military and Diplomacy.

We are working with domain experts to formally evaluate the learned concept hierarchy. A political scientist (personal communication) comments:

> The international affairs topic does an excellent job of capturing the key distinction between military/defense and diplomacy/aid. Even more impressive is that it then also captures the major policy areas within each of these issues: the distinction between traditional military issues and terrorism-related issues, and the distinction between thematic policy (e.g., human rights) and geographic/regional policy.

## 5    Conclusion

We have presented L2H, a model that discovers not just the interaction between overt labels and the latent topics used in a corpus, but also how they fit together in a hierarchy. Hierarchies are a natural way to organize information, and combining labels with a hierarchy provides a mechanism for integrating user knowledge and data-driven summaries in a single, consistent structure. Our experiments show that L2H yields interpretable label/topic structures, that it can substantially improve model perplexity compared to baseline approaches, and that it improves performance on a multi-label prediction task.

## Acknowledgments

We thank Kristina Miler, Ke Zhai, Leo Claudino, and He He for helpful discussions, and thank the anonymous reviewers for insightful comments. This research was supported in part by NSF under grant #1211153 (Resnik) and #1018625 (Boyd-Graber and Resnik). Any opinions, findings, conclusions, or recommendations expressed here are those of the authors and do not necessarily reflect the view of the sponsor.

## Footnotes

[3]In this paper, we use *node* when emphasizing the structure discovered by the model. Each node corresponds to a *concept* which consists of a *label* and a *topic*.

[4]We find bigram candidates that occur at least ten times in the training set and use a $\chi^2$ test to filter out those having a $\chi^2$ value less than 5.0. We then treat selected bigrams as single word types in the vocabulary.

[5] We run L-LDA on train for 1,000 iterations and ten models after 500 burn-in iterations. For each model, we sample assignments for all tokens using 100 iterations and average over chains to estimate $\hat{\theta}_d^{\text{L-LDA}}$.

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
