[Reviews · NeurIPS 2014]

Submitted by Assigned_Reviewer_2

This paper proposes a version of labeled LDA that (1) supports multi-labeled data; (2) produces a label hierarchy. Comparisons with baselines on congressional bills shows improvements in perplexity as well as classification accuracy.

The approach is a natural way to handle multi-labeled data, and the hierarchy inference approach appears sound. The paper is generally well-written, and the baseline comparisons are sufficient.

I have a few suggestions / concerns:

- It is not clear why the topic assignments are then used as features in an SVM. Couldn't we do MAP inference directly to estimate L_d^1 ?

- The switching variable \pi_d seems strange. It this a result of restricting each tag to have exactly one topic? Please better motivate and clarify. What happens if we do not include the switching variable? What problem is introduced?

- Is comparing the classification accuracy of LDA really fair here, since L2H uses internal nodes (and thus has more features)?

Smaller comments to improve clarity:

- Please define C_{d,i} earlier.

- Eq 3 might be better placed near the start of Section 3.

Post rebuttal:

The authors have mostly addressed my concerns, and also fixed an evaluation bug pointed out by another reviewer.
Summary: This paper proposes a natural extension to existing supervised topic models to support multi-labeled data and produce hierarchies of labels. While mostly a combination of existing ideas, I feel the paper does provided advances in this well-studied area.

Submitted by Assigned_Reviewer_18

Summary: In this paper the authors proposed a model to learn a concept hierarchy from multi-labeled documents. The proposed model is principled, elegant and sound. The evaluation was done on a political data set where the labels are political issues. While it would have been better if the authors had tried using more corpora to evaluate the model, overall the paper is clear and the method is novel.

Quality: The paper studies a particular way of constructing a concept hierarchy that does not seem to have been well studied before. I suppose this special setup has many real-world applications, but it would be nice if the authors could provide more actual examples where this kind of concept hierarchy is desirable. The proposed model is clearly presented. The design choices are either intuitive or well justified, so it is not hard to follow the paper. The evaluation is also clearly presented. It would be nicer if more data sets from other domains could be used in the evaluation, since after all the problem studied is fairly general.

Clarify: I find the presentation quite clear.

Originality: To the best of my knowledge the presented model is different from models I’m aware of.

Significance: The proposed model may inspire more work along this direction.
Summary: This paper nicely presents a new model for constructing a concept hierarchy from multi-labeled documents. The model is principled and elegant. The evaluation also shows its advantage.

Submitted by Assigned_Reviewer_19

Given a corpus of multi-labeled documents, the author(s) propose a topic model which associates each label with a topic, and recovers a tree-structured hierarchical taxonomy of the topics/labels. The model is evaluated on a political science data set. As well as finding such interpretable structure, as seen in qualitative results in the paper, the model is reported to improve perplexity over LDA and labeled LDA, and also provides features which outperform TF-IDF and labeled LDA at multi-label document classification.

Using label information to create a hierarchy of topics and labels is a nice idea, bringing together separate strands of work on hierarchies of topics and hierarchies of labels. The proposed model is elegant and is a principled approach to the problem. An approximation must be made in the Gibbs sampling procedure (Equation 7), though it is unclear how accurate the approximation is. Although the procedure of label selection using TF-IDF when performing document classification seems somewhat ad-hoc, to be fair it is a reasonable approach, and the strategy has been used before by Ramage et al.

However, unfortunately, the procedure used to estimate the perplexity scores for Equation 9 is incorrect. In the following, for simplicity of exposition, I will describe the issue for standard LDA instead of for the proposed model, ignoring the variables which are specific to this model such as the labels. The notation here also explicitly conditions on \alpha.

In the manuscript, for each MCMC sample of the topics, the procedure seems to be to draw Z from its posterior given each test document. This procedure fits Z to the *test* data. It would therefore over-estimate the likelihood of data which are truly unobserved, and would reward over-fitting of Z.

In more detail, the procedure in the manuscript estimates \sum_z P(W|Z, \phi, \alpha) P(Z|W, \phi, \alpha). However, what should be estimated for Equation 9 is, for each sample of \phi, the likelihood for the test document, P(W|\phi, \alpha) = \sum_z P(W,Z|\phi, \alpha) = \sum_z P(W|Z, \phi) P(Z|\alpha). In other words, the present procedure computes an average over the *posterior*. However, the likelihood is equal to an average over the *prior*.

To correct this, an unbiased estimate for the likelihood, to plug into Equation 9, can be computed using these samples from the posterior, by instead taking the harmonic mean instead of the arithmetic mean (Equation 15 in Wallach et al. 2009). However, Wallach et al. found that this method has extremely high variance, and hence is not recommended. They suggest alternative algorithms such as the left-to-right method and annealed importance sampling.

Another sensible alternative strategy is to repeat the exact procedure used in the manuscript, except first splitting each test document into two portions, "test_train" and "test_test." The procedure in the paper (sample Z from the posterior given the data and topics) is performed on "test_train," and the Rao-Blackwellized estimates of theta from the resulting Z's are used to evaluate perplexity on "test_test," thus computing P(test_test|test_train, \phi). This is the "estimated theta document completion" strategy, described in Section 5.1 of Wallach et al. (2009). For more information on evaluating topic models, see:

Wallach, H. M., Murray, I., Salakhutdinov, R., & Mimno, D. (2009). "Evaluation methods for topic models." Proceedings of the 26th Annual International Conference on Machine Learning.

Going forward, it would be good if the author(s) could comment on the semantics of the extracted hierarchy. Clearly the model does not always place strictly more specific topics/labels below more general topics/labels, although the generative model encourages this somewhat. So if the tree structure does not always encode generality relationships, how should we interpret it? It would also be good to present results on more than one data set to give more evidence that the results hold generally.

*** Thanks to the authors for addressing my concern regarding the evaluation procedure. I have adjusted my score accordingly.
Summary: The paper proposes an elegant topic model for finding tree-structured label hierarchies, with topics associated with the labels. A key evaluation step was previously incorrect, but the authors have posted corrected results in their response.
Author Feedback
Author rebuttal: We thank the reviewers for their helpful comments and suggestions.

We especially thank Reviewer_19 for pointing out in details the procedure for computing held-out likelihood. We indeed made a mistake on the likelihood evaluation. After computing the perplexity again using the "estimated theta document completion" strategy (Section 5.1, Wallach et al. 2009), the new mean perplexities averaged over 5 folds are as follows:
109 110 111 112
LDA 158 179 200 220
L-LDA 215 220 272 302
L2F 219 219 271 297
L2H 168 183 237 247

The new results show that LDA does much better than the other models with labels since it can freely optimize the likelihood without additional constraints. However, this does show that when using labels (which improve classification and interpretability), learning the topic hierarchy improves the generalizability of Labeled LDA.

Reviewer_19 also comments about the semantics of the learned hierarchy and the relationships between nodes. Our model tries to learn these relationships by encouraging label-A to be the parent node of label-B when (a) label-A and label-B co-occur frequently and label-A is more popular (i.e., using label co-occurrences), and (b) the words in documents tagged with label-A are more general than in documents tagged with label-B (i.e., word usages). So it is possible that in some datasets that a more specific label, e.g., Higher Education, might be placed as the parent node of a more general label, e.g., Education, if the dataset contains a lot of documents about Higher Education but only part of these documents are also tagged with Education. To correct for this problem, we might need to use additional resources in future work.

For Reviewer_2's questions:

- It is not clear why the topic assignments are then used as features in an SVM. Couldn't we do MAP inference directly to estimate L_d^1 ?

Right, we tried to use MAP inference to estimate L_d^1 directly but the performance was not great. This is a similar problem with Labeled-LDA and as Reviewer_19 also pointed out, Ramage et al also used TF-IDF when performing classification

- The switching variable \pi_d seems strange. It this a result of restricting each tag to have exactly one topic? Please better motivate and clarify. What happens if we do not include the switching variable? What problem is introduced?

The main reason for using \pi_d is that it provides a natural way to compute the probability of assigning a token to a node in L_d^0. This is particularly useful when we need to update the tree structure: when a new tree structure is proposed, some tokens which were in L_d^1 will become in L_d^0 (and vice versa), so we need to be able to compute the probability of both cases.

- Is comparing the classification accuracy of LDA really fair here, since L2H uses internal nodes (and thus has more features)?
To be more precise, we use LDA as a baseline for held-out likelihood, which is common practice in the topic modeling community.